# Digital twin based monitoring and control for DC-DC converters

Zhongcheng Lei [1], Hong Zhou[1], Xiaoran Dai [1]✉, Wenshan Hu [1]✉ & Guo-Ping Liu [2]

The monitoring and control of DC-DC converters have become key issues since DC-DC converters are gradually playing increasingly crucial roles in power electronics applications such as electric vehicles and renewable energy systems. As an emerging and transforming technology, the digital twin, which is a dynamic virtual replica of a physical system, can potentially provide solutions for the monitoring and control of DC-DC converters. This work discusses the design and implementation of the digital twin DC-DC converter in detail. The key features of the physical and twin systems are outlined, and the control architecture is provided. To verify the effectiveness of the proposed digital twin method, four possible cases that may occur during the practical control scenarios of DC-DC converter applications are discussed. Simulations and experimental verification are conducted, showing that the digital twin can dynamically track the physical DC-DC converter, detect the failure of the physical controller and replace it in real time.

With the rapid advances in power electronics, DC-DC converters have become a crucial part of electric vehicles, DC microgrids, and renewable energy generation systems, which have been widely used in these areas and applications[1]. As illustrated in Fig. 1, the DC-DC converter plays a vital role in renewable energy systems, which can provide feasible voltage through the stepping up/down method for various loads with different voltage specifications. Considering the crucial role of DC-DC converters in power electronics applications, the control and monitoring of the status of DC-DC converters is of great importance[2], and is also challenging due to intrinsic nonlinear characteristics[3], therefore, the system state should be estimated online[4]. Various research studies have been conducted to address the nonlinear issues of nonlinear circuits, for example, fuzzy observer-based state estimation based on a slack variables technique[5] and a switching multi-instant fuzzy observer to obtain less conservative results[6]. A recent study enlarged the application scope of fuzzy state estimation in nonlinear circuits with better estimation performance[4].

The digital twin (DT) is an important technology that can be employed in different industrial applications. Currently, the DT has become an increasingly popular research topic, with related research mainly focusing on smart manufacturing and Industry 4.0[7–9], web-based thermal power plants[10], and driver assistance systems[11,12]. Digitized laboratories[13], which are a kind of DT for laboratory setups, have also become a kind of DT online laboratory in recent years[14,15].

As a transforming technology, the DT is one of the most studied topics within the power electronics area[16,17], and is mainly focused on condition and health monitoring. In Ref. 16, a survey was carried out to identify the concepts, applications, challenges, and trends of DTs used in energy conversion systems based on power electronics. In Ref. 17, a DT-based health indicator estimation method for the condition monitoring of DC-DC converters was proposed, which is mainly used for monitoring the degradation level of the capacitor and MOSFET (metal-oxide-semiconductor field-effect transistor), the control perspective of which has not been considered. The application of DTs in DC-DC converters can potentially provide benefits such as health monitoring, fault diagnosing[18] in photovoltaic systems and detection of system abnormalities for buck converters[19].

In this work, a DT-based buck converter system is explored for simplicity without loss of generality, mainly from control perspectives.

[1]Department of Artificial Intelligence and Automation, School of Electrical Engineering and Automation, Wuhan University, Wuhan 430072, China. [2]Center for Control Science and Technology, Southern University of Science and Technology, Shenzhen 518055, China. ✉e-mail: dai_xiaoran@whu.edu.cn; wenshan.hu@whu.edu.cn

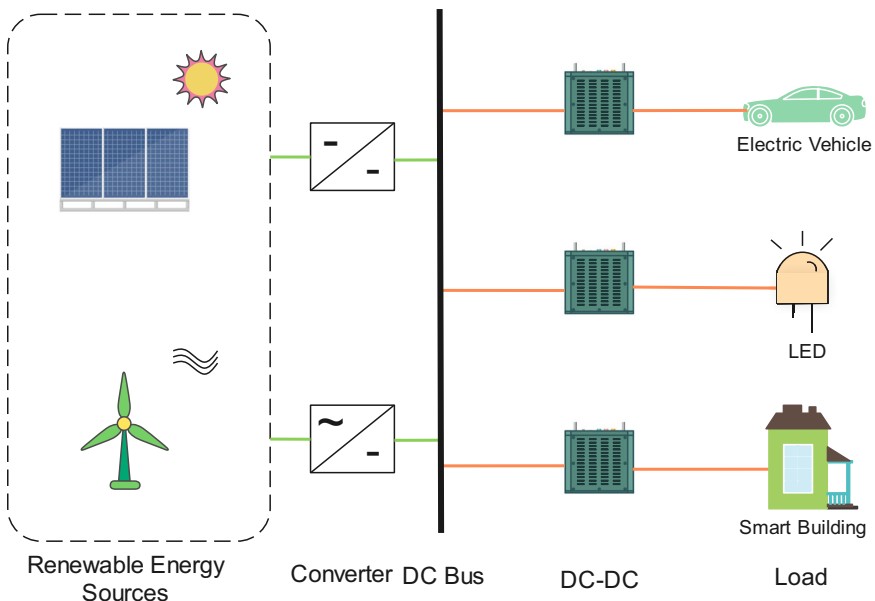

**Fig. 1 | Schematic of renewable energy systems, in which the DC-DC converter plays a vital role.**

The proposed DT system includes both the hardware of the buck converter system (physical part) and the DT system (digital part) running in RT-LAB instead of a hardware-in-the-loop (HIL) approach that merely emulates the converter systems in RT-LAB[20]. The proposed DT system is proven to be effective for four possible cases that may occur during practical control scenarios: 1) tracking the physical system and ensuring dynamic characteristics when the reference value changes; 2) automatically adapting to the system model variation to achieve good control performance when the input voltage varies; 3) replacing the failed controller in the physical converter system with a DT controller; and 4) suiting physical DC-DC converters with different components, such as insulated gate bipolar transistors (IGBTs) and silicon carbide (SiC) MOSFETs.

## Results

### Simulation setup

To demonstrate the effectiveness of the DC-DC converter DT modeling method, four simulation cases are conducted as discussed in the following subsections. The parameters for the simulation are listed in Table 1, in which 48 V is a type of low-voltage dc microgrids that has been widely used for residential and communication applications[21,22] and for simulation and experimental verification in microgrids and renewable energy studies[23,24]. The time step employed for running the DT is 0.1 ms.

**Table 1 | Parameters for the DT experiments**

| Parameter name | Value | Description |
|---|---|---|
| DC input voltage ($V_{in}$) | 100 V → 90 V → 80 V | |
| IGBT switching frequency | 10 kHz | For simulation and experimental verification in Cases I to III |
| SiC MOSFET switching frequency | 100 kHz | For simulation and experimental verification in Case IV |
| Inductor ($L$) | 2 mH | |
| Filter capacitance ($C$) | 3300 μF | |
| Load resistance ($R$) | 20 Ω | |
| Voltage reference | 48 V | The rated voltage for low voltage dc microgrids |

### Simulation case I: reference value tuning

In industry, especially in energy-related applications, voltage reference changes are commonly employed to evaluate the tracking capability of a proposed system[25,26]. Moreover, in the context of control-oriented research studies, it is prevalent to perform voltage reference changes, typically in the form of step-up or step-down adjustments, to assess the tracking performance[27,28]. Therefore, reference tracking, as a basic control task, is first verified in this case.

Although 48 V is considered in this work, different output voltages may be needed in other scenarios. For example, 20 V was used for charging batteries[29]. Fig. 2a shows the simulation results when the reference voltage is tuned from 0 V to 48 V and then from 48 V to 20 V. It can be seen that the DT system can track the physical buck converter system, while the mechanism model-based system cannot track the transient state.

### Simulation case II: system model variation

For a renewable system, due to the intermittent feature, the input voltage may vary from time to time. This case emulates a scenario in which the input voltage of a renewable system varies. From the subsequent modeling process, it can be concluded that once the input voltage changes, the control matrix $\begin{bmatrix} \frac{V_{in}}{L} \\ 0 \end{bmatrix}$ will change, thus, the traditional mechanism model that is built based on a fixed input voltage $V_{in}$ is no longer accurate.

Figure 2b shows the simulation results of the output voltage when the input voltage is changed from 100 V to 90 V, and then from 90 V to 80 V, in which the DT system can track the physical system better than the mechanism model-based system.

### Simulation case III: DT responses under controller failure

For practical applications, controller failure that is frequently encountered due to various reasons can cause system malfunctioning[30], and may cause failures and damage to the entire system, which should be seriously considered and addressed to ensure reliable control. For power electronics applications, controller failure poses a significant challenge. For a photovoltaic power plant, down (not functional) probability is predominantly caused by controller and inverter failures[31]. Power converter and control circuit defects, such as sensor failure and controller failure, contribute to around 90% of electric drive failures[32,33].

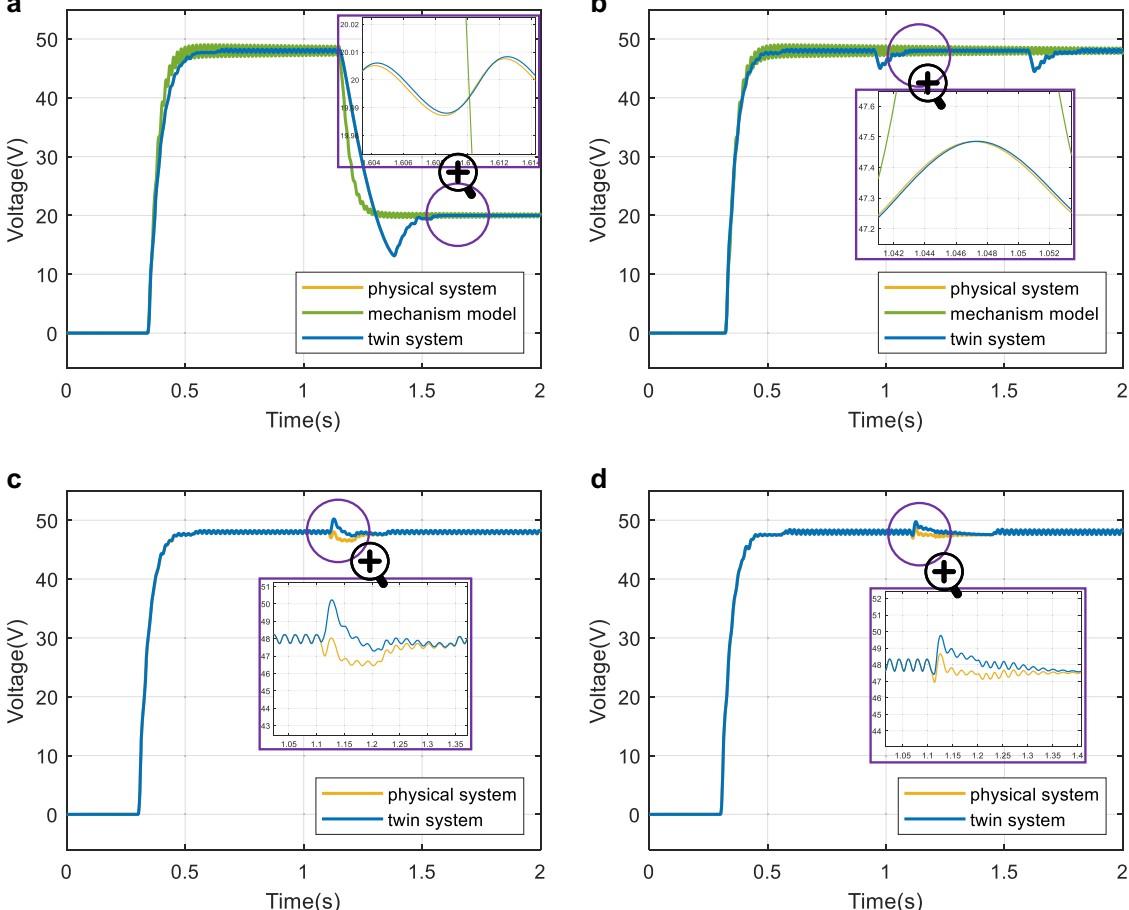

**Fig. 2 | Simulation results of the four cases. a** The output voltage when the reference voltage is changed from 0 V to 48 V and from 48 V to 20 V. **b** The output voltage when the input voltage is changed from 100 V to 90 V, and then from 90 V to 80 V. **c** The output voltage when controller failure is detected and replaced by the DT controller to ensure a functioning system. **d** SiC MOSFET is used as the switch in the circuit as the case in (**c**). Source data are provided as a Source Data file.

Figure 2c shows the simulation results of the output voltage when controller failure is detected and replaced by the DT controller. It can be concluded that the DT controller can ensure a functioning system when the physical controller fails.

**Simulation case IV: generalization ability verification**
SiC MOSFETs have a lower internal resistance and higher switching frequency than IGBTs, thus, they have been gradually applied in DC-DC converters[34,35]. To illustrate the generalization ability of the proposed method with different devices and a wide range of switching frequencies, in this case, the IGBT in the previous three cases has been replaced by an SiC MOSFET based on Ref. 36. The simulation result in Fig. 2d shows that it can achieve good control performance and adapt quickly in the case of controller failure and replacement.

**Experimental setup**
Experimental verification is conducted with respect to the four cases presented in previous simulation subsections to verify the effectiveness of the proposed DT system. Fig. 3 shows the experimental setup for the verification, in which the upper part is the RT-LAB OP 5700, including the front and back views. The parameters for the DT experiments are the same as the simulation, as listed in Table 1. The time step used for running the DT is 0.1 ms, which is aligned with the simulation time step. The physical system is a buck converter that is controlled by a digital signal processor (DSP) TMS320F28335, while the dynamic model executed in RT-LAB works as the DT. Other hardware includes the power source IT-M3123, the IGBT intelligent power module PM50RLA120 (for experimental verification of Cases I to III), and the SiC MOSFET NTH4L040N120SC1 (for experimental verification of Case IV). The mechanism model is also run in RT-LAB to provide a reference for comparison with the DT model. As the changes cannot be seen from the outside, the three-dimensional model is not necessarily required for DT implementation.

**Experimental case I: reference value tuning**
This experimental verification corresponds to Simulation Case I for reference tracking. Figure 4 illustrates the experimental results when the reference voltage is changed from 0 V to 48 V and then from 48 V to 20 V. The DT system can track the physical buck converter system much better than the mechanism model-based system. Specifically, the twin system is observed to track changes more effectively in cases of abrupt reference variations, whereas the model-based system shows more oscillations during dynamic processes due to parasitic components in the circuit that were not considered during the modeling process. In contrast, the DT technique is capable of mitigating the adverse effects of these parasitic components. In some special scenarios, the dynamic characteristics are also worth considering[1], which is also the advantage of the DT model compared to the mechanism model.

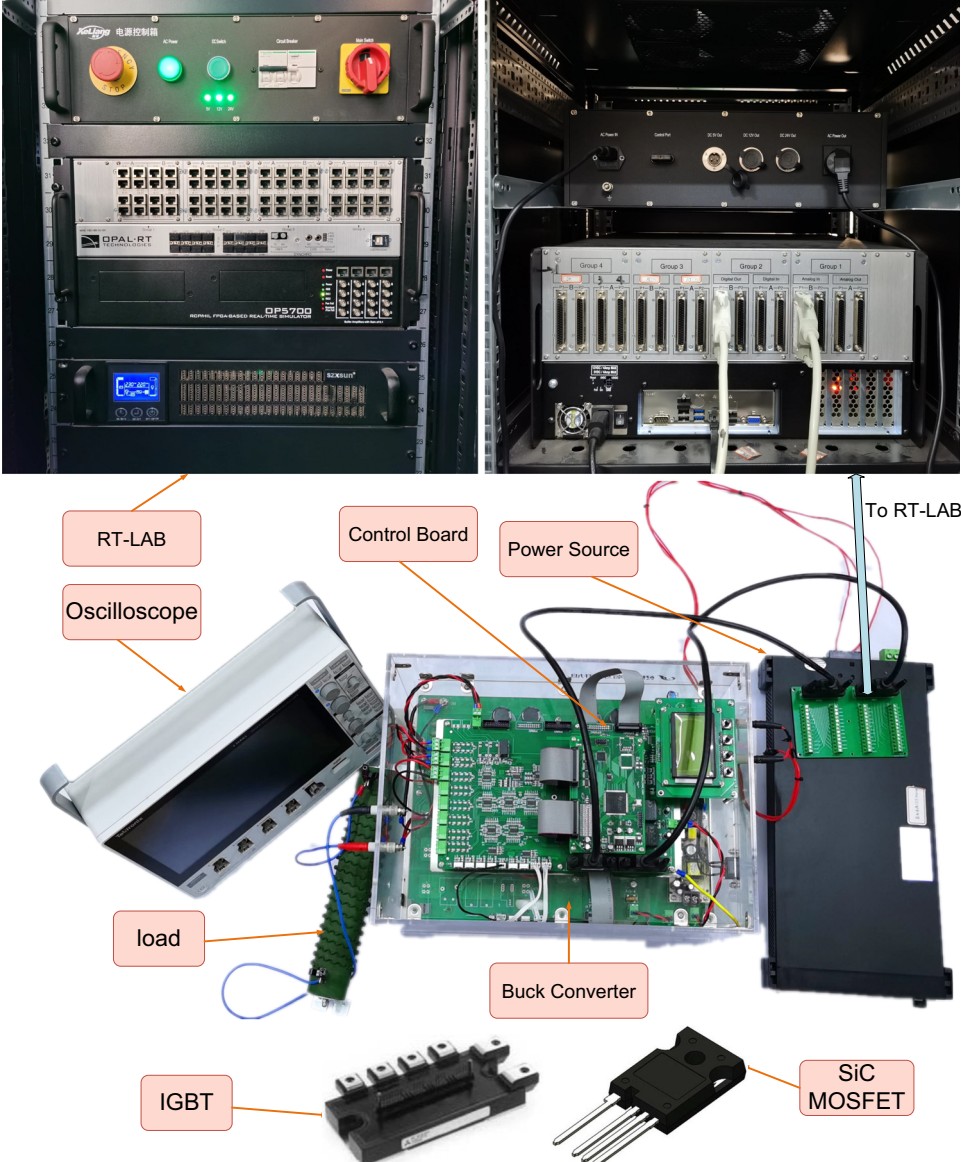

**Fig. 3 | Experimental setup for the DT buck converter experiment.** The IGBT and SiC MOSFET used are also included.

## Experimental case II: system model variation

The purpose of this experiment is to verify the ability of the DT model to adapt to variations in the system model, which corresponds to Simulation Case II. For the experiment in Fig. 5, the mechanism model has no knowledge of the variance of the input voltage; therefore, its control output remains the same (see the duty cycle in the upper part of Fig. 5), which will affect the output voltage; thus, the mechanism model is not applicable. Observations of the converter's duty cycle reveal that while the duty cycle of the twin system varies with changes in input voltage, the mechanism model remains unchanged and introduces errors even in steady-state conditions. However, the DT model can automatically adapt to the system model variation and adjust the control output $\mu$ to track the physical system (see the output voltage in the lower part of Fig. 5), and finally achieve good control performance, as shown in Fig. 5. This capability of the twin model to closely track variations in the system model proves advantageous for monitoring purposes and facilitates controller replacement.

## Experimental case III: DT responses under controller failure

As shown in Fig. 6, once the control input $\mu = 0$ has been detected, which means that the physical controller of the physical buck converter system fails, the DT controller that is constructed based on hybrid twin modeling and runs in the RT-LAB will be activated in a very short time and works as the controller to ensure a functioning system. The DT controller can be regarded as a redundant system to improve the reliability and stability of the physical system. As a cost-effective option, once controller failure has been detected and the RT-LAB that works as a DT controller is activated, a new DSP controller can be used to replace the failed controller at appropriate timing.

Although direct comparison between methods can be challenging, the proposed method was compared to previous methods from three aspects: whether a physical DC-DC converter is involved, generalization ability, and applicable cases (three cases discussed in the previous sections). Table 2 shows the comparison results. The proposed method can be used for more scenarios for the monitoring and control of physical DC-DC converters compared with HIL[20], DT-based monitoring[37] and Internet-of-Thing based monitoring and control[38].

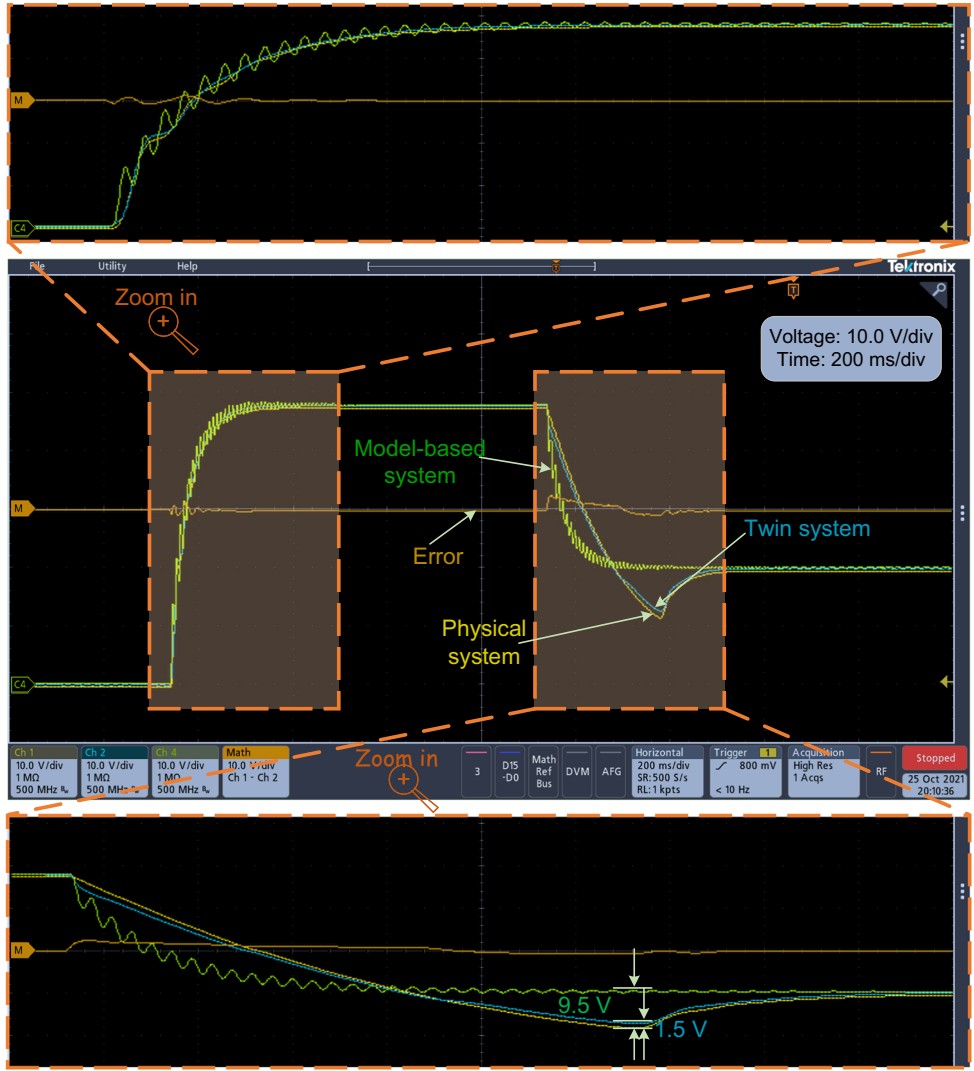

**Fig. 4 |** Experimental results of the output voltage when the reference voltage is changed from 0 V to 48 V and from 48 V to 20 V.

**Experimental case IV: generalization ability verification**

To further verify the generalization ability of the proposed system and validate the results obtained from the simulation case in Simulation Case IV, experimental verification regarding the SiC MOSFET is also conducted with a new set of buck converter. Owing to the adoption of the SiC MOSFET, the new converter has a compact size compared to the IGBT-based counterpart, maintaining identical voltage and power levels.

The experimental results of the output voltage are shown in Fig. 7. In this case, the duty cycle is the same as that in Experimental Case III. The transient state is observed as shown at the top of Fig. 7, which demonstrates a remarkable dynamic tracking performance. For controller failure, the results show that the system can quickly adapt to the DT controller, thus, ensuring a smooth transition and maintaining good control performance. It can be seen that the error of the output voltage between the physical system and the DT system is 1.0 V, which is lower than 1.6 V in Fig. 6.

## Discussion

In summary, in this work, methodologies to achieve DT based monitoring and control of DC-DC converters have been explored. A DT DC-DC converter has been designed and implemented that employs a hybrid twin modeling method. In total, four cases were investigated. Three cases with reference value tuning, system model variation, and

controller failure that may occur during practical control scenarios were considered and demonstrated with simulation and experimental verification. The experimental results are in high agreement with the simulation results and are consistent with the theoretical analysis, which further validates the effectiveness of the proposed method. Simulation and experimental verification cases with an SiC MOSFET are also conducted and verify the generalization ability of the proposed method. The results show that dynamic system synchronization is achieved where the DT can dynamically track the physical system even when the system model varies, and controller failure can be detected and replaced, which verifies the effectiveness of the proposed DT method for monitoring and control. In future work, the DT could be used to address possible sensor failure, system condition diagnosis and prognosis, and optimal control. DT controller optimization is also a challenging and promising direction for future work.

## Methods

In this section, the design and implementation of the DT DC-DC converter based on the physical DC-DC converter are introduced.

### Physical and twining system of the DC-DC converter

A typical DC-DC buck circuit is illustrated in Fig. 8a, along with its equivalent circuit during the ON and OFF states. Within one switching cycle of the switch, the system dynamic equations can be derived

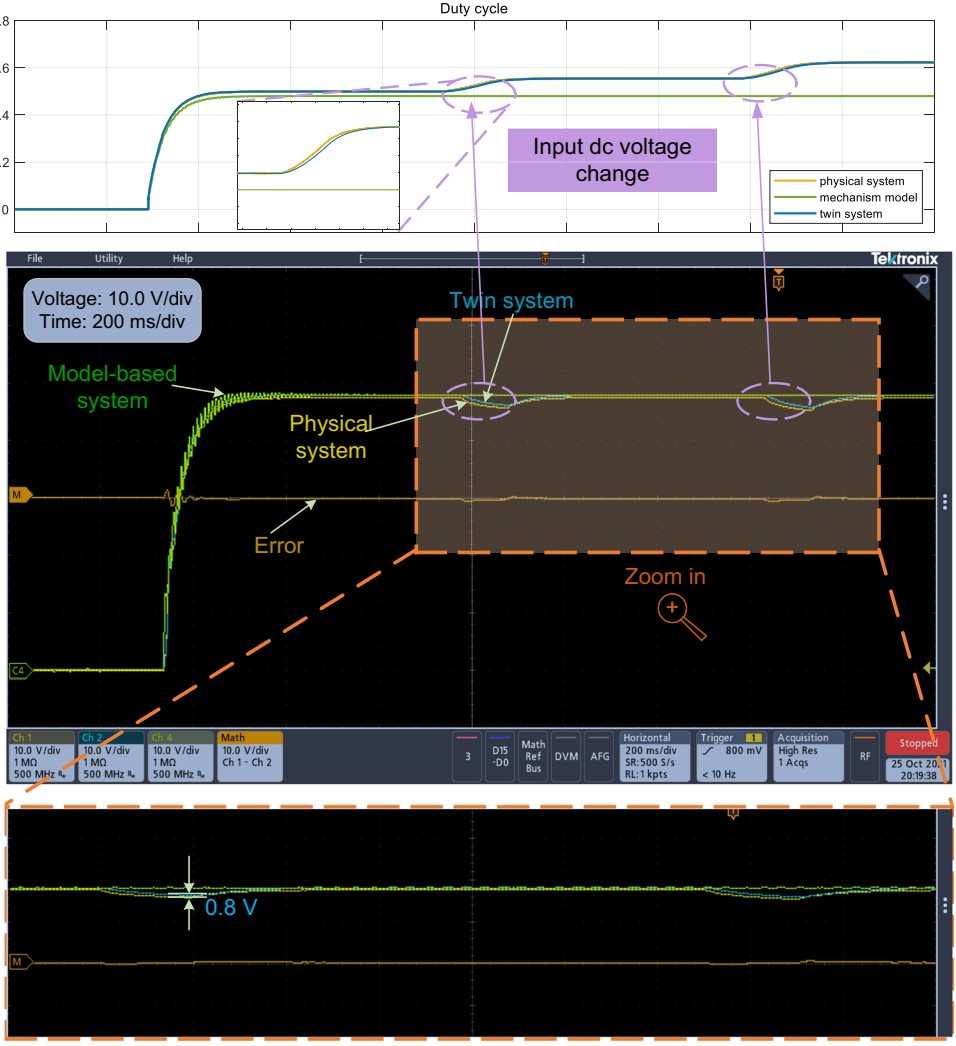

**Fig. 5 | Experimental results of the output voltage and the corresponding duty cycle when the input voltage is changed from 100 V to 90 V, and then from 90 V to 80 V.** Duty cycle source data are provided as a Source Data file.

based on Kirchhoff's laws[39]. During the ON state of the switch, the inductor current $i_L$ and the capacitor voltage $u_C$ can be expressed as follows:

$$i_L = C\frac{du_C}{dt} + \frac{u_C}{R},$$
$$u_C = V_{in} - L\frac{di_L}{dt}, \tag{1}$$

where $V_{in}$ represents the dc source input (not a constant for renewable energy systems due to the intermittent feature), and $L$, $C$, and $R$ represent the inductance of the inductor, the capacitance of the capacitor, and the load resistance, respectively. During the OFF state of the switch, their state equations are given by

$$i_L = C\frac{du_C}{dt} + \frac{u_C}{R},$$
$$u_C = -L\frac{di_L}{dt}. \tag{2}$$

Let the duty cycle be defined as $\mu \in (0,1)$, which is also the control input. Using the state-space averaging method, the unified state equation expression for the buck circuit over the entire switching cycle

is obtained as

$$\begin{bmatrix} \dot{i}_L \\ \dot{u}_C \end{bmatrix} = \begin{bmatrix} 0 & -\frac{1}{L} \\ \frac{1}{C} & -\frac{1}{RC} \end{bmatrix} \begin{bmatrix} i_L \\ u_C \end{bmatrix} + \begin{bmatrix} \frac{V_{in}}{L} \\ 0 \end{bmatrix} \mu. \tag{3}$$

Figure 8b demonstrates the physical DC-DC converter and its DT counterpart. As a DT DC-DC converter, the two key features of the physical and twin systems are as follows.

1) Data transmission and dynamic DT: The physical system runs in real-time, while the DT can dynamically track the physical system. In this scenario, the physical system sends the data including $i_L$, $u_C$, and $\mu$ to its digital counterpart, based on which the DT updates its model and parameters. This also suits for variations of the system model. For a mechanism model that is built based on a fixed system offline, the model can be inaccurate once the system parameters change. While for a DT model in this work, it is updated online according to the real-time data from the physical system. Synchronization between the DT and the physical system can be ensured even if the physical model changes or parameters drift due to aging. 2) Information exchange and update: Since the DT monitors the physical system, once the controller failure of the physical system has been detected, the DT controller can be switched to control the physical system to ensure a functioning system.

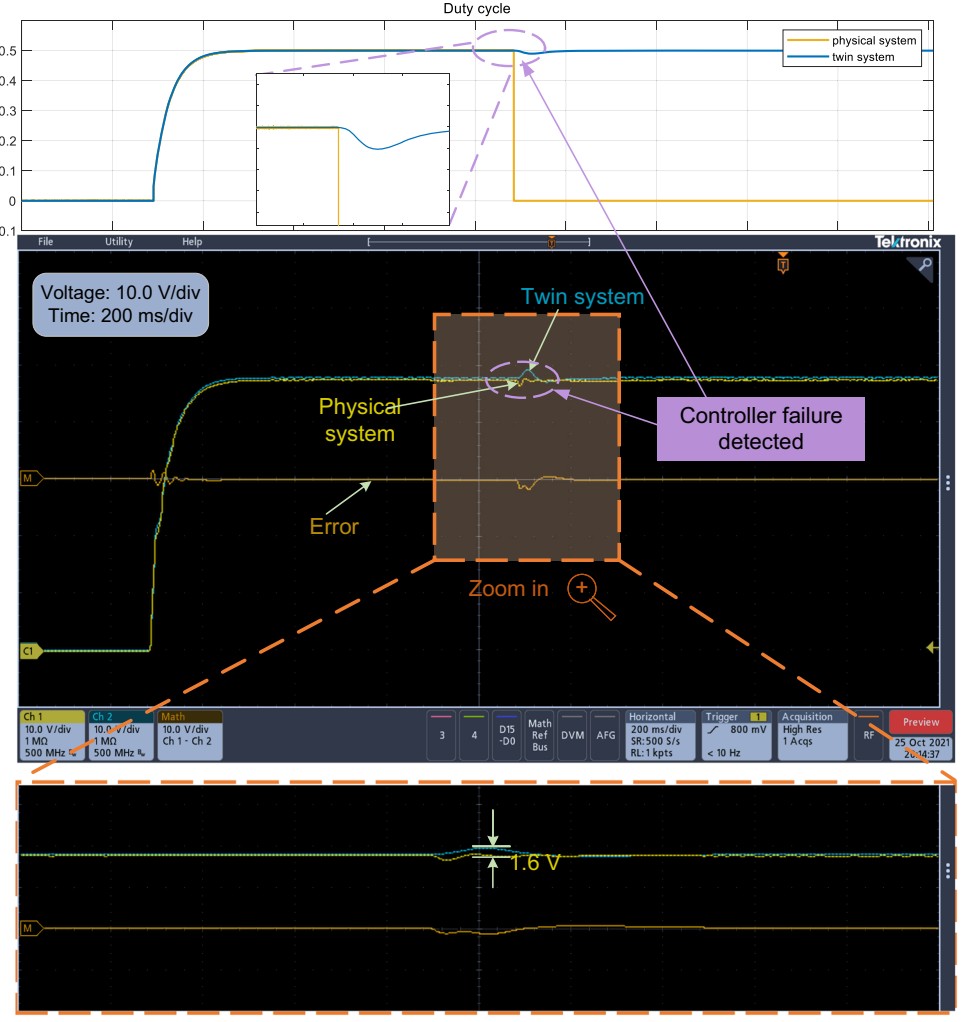

**Fig. 6 | Experimental results of the output voltage and the corresponding duty cycle when physical DSP controller failure is detected and has been replaced by the DT controller to ensure a functioning system.** Duty cycle source data are provided as a Source Data file.

Based on circuit theory, the process knowledge/mechanism modeling can be used. This model is easy to acquire and comprehend with actual measured values. However, it is generally fixed and thus suffers from parameter drift issues, which means that once the system model of the circuit system changes, the model has to be reconstructed from scratch. For example, $V_{in}$ is typically fixed when using mechanism-based modeling owing to cost considerations. In addition, in renewable energy systems, voltage fluctuations (changes of $V_{in}$) are common due to variability on the generation side. Even if $V_{in}$ is not fixed and measurable, the $R$, $L$ and $C$ components within a specific DC-DC system may be time-varying, which makes mechanism-based modeling challenging. To solve this issue, a mechanism-data hybrid modeling method is adopted for the design and implementation of the DT model.

To realize the hybrid twin modeling, system (3) is discretized to the form of (4), where $y$ and $\hat{y}$ are the observed output and estimated value, respectively. $\theta$ and $\hat{\theta}$ are the parameter matrix and estimated parameter matrix, respectively. $\psi$ is the input vector. $A_d$ and $B_d$ (noting that $V_{in}$ is included in $B_d$) are the system and control matrices discretized from (3). It is crucial to emphasize that (4) represents a time-variant system, capable of accurately reflecting the parameters of the physical system. This indicates that the proposed system remains applicable and effective, even in the presence of parameter drift, such as degradation in

capacitors or MOSFETs.

$$\underbrace{\begin{bmatrix} i_L(t+1) \\ u_C(t+1) \end{bmatrix}}_{y(\hat{t})} = \underbrace{\begin{bmatrix} A_d | B_d \end{bmatrix}}_{\theta(t)} \underbrace{\begin{bmatrix} i_L(t) \\ u_C(t) \\ u(t) \end{bmatrix}}_{\psi^T(t)} \quad (4)$$

Different approaches can be employed for state estimation, for example, Kalman filters and observers, however, it has been proven that the Kalman filter provides better estimation performance for states and exhibits satisfactory estimation accuracy in the presence of noise[40]. The DT is connected with the physical system, which contains

**Table 2 | Proposed Method Compared With Previous Methods**

| Method | Proposed | 20 | 37 | 38 |
|---|---|---|---|---|
| Physical DC-DC converters involved | ✓ | | ✓ | ✓ |
| Generalization ability | ✓ | ✓ | ✓ | |
| Reference tracking | ✓ | ✓ | ✓ | ✓ |
| System variation | ✓ | ✓ | | ✓ |
| Controller replacement | ✓ | | | |

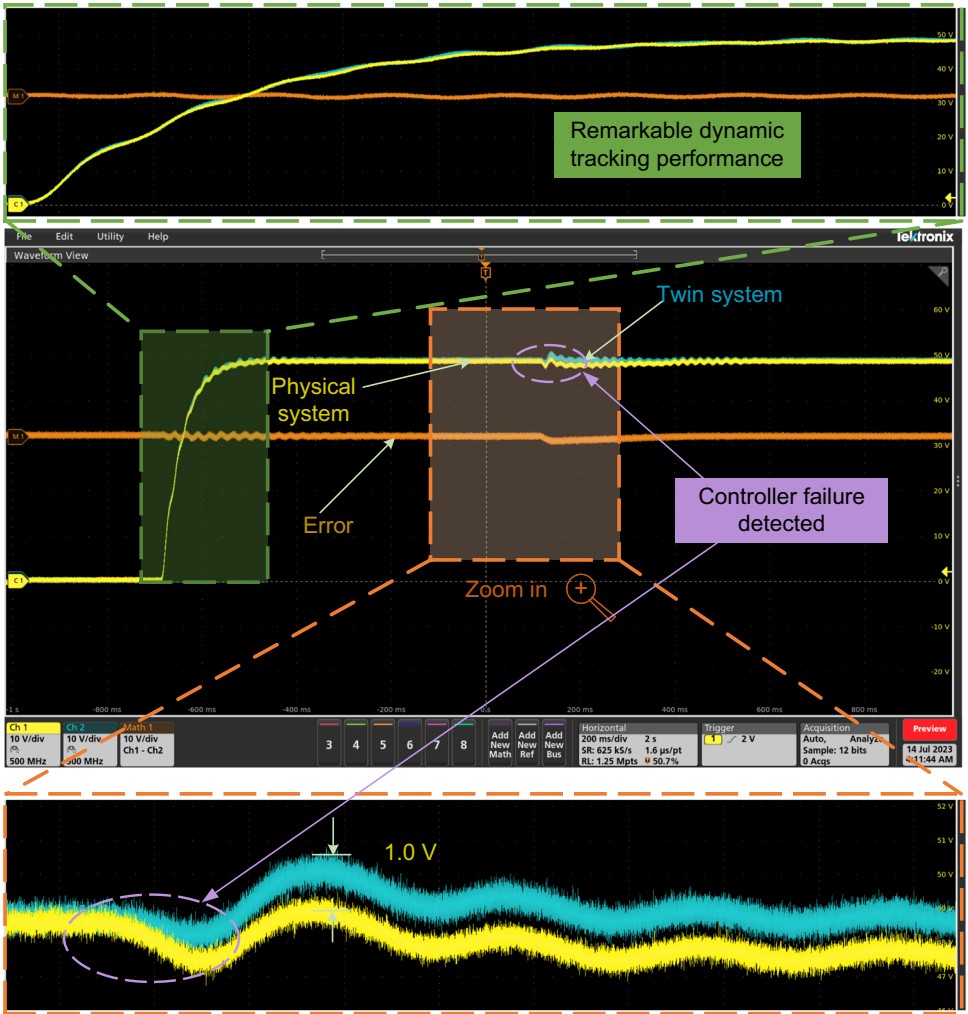

**Fig. 7 |** Experimental results of the output voltage when physical DSP controller failure is detected and has been replaced by the DT controller to ensure a functioning system (using SiC MOSFET as the switch).

considerable noise, therefore, traditional parameter estimation methods are not effective. The Kalman filter adaptation algorithm[41], which possesses several advantages, such as the ability to reduce noise, low computational complexity, good convergence[42], and high time efficiency, can be used to address this issue:

$$\hat{\theta}(t) = \hat{\theta}(t-1) + K(t)\big(y(t) - \hat{y}(t)\big) \quad (5)$$

$$K(t) = Q(t)\psi(t) \quad (6)$$

$$Q(t) = \frac{P(t-1)}{R_2 + \psi^T(t)P(t-1)\psi(t)} \quad (7)$$

$$P(t) = P(t-1) + R_1 - \frac{P(t-1)\psi(t)\psi^T(t)P(t-1)}{R_2 + \psi^T(t)P(t-1)\psi(t)} \quad (8)$$

Estimation (5) can be employed to update $\theta$ in real time, where the gain $K$ can be designed by a Kalman-filter-based recursive estimation algorithm and calculated using (6). As $V_{in}$ is included in $B_d$ (thus in $\theta$), it is also updated in real time. $Q(t)$ can be derived from (7) and (8), where $R_1$ is the covariance matrix of the Gaussian white noise of true parameters and $R_2$ is the variance of the noise source. The computational cost of the Kalman-filter-based

estimation is closely linked to the number of estimated state variables[43]. For the case of the model with two estimated variables in this work, the computational cost is relatively low. For complex nonlinear models and systems with more state variables, the feedback linearization technique can be an effective solution for reducing complexity[1].

It should be noted that the Kalman filter is only applied to the DT model, while the parameters in the mechanism-based model are actual measured values. The aim of the mechanism-based model is to accurately represent the fundamental physical principles and dynamics of the DC-DC converter. The DT model, on the other hand, facilitates real-time analysis, monitoring, and control through live data, making the Kalman filter an advantageous choice due to its capacity to enhance state estimation accuracy and enable effective control.

## Control architecture
Fig. 8c illustrates the control architecture of the physical-DT DC-DC converter, which includes two closed loops, one for the physical system and another for the DT system. The control architecture is based on the two aforementioned key features of the physical and twin systems, namely, the DT model and its parameters are updated in real-time based on the physical system, and the DT controller can be used to control the physical system if physical controller failure occurs.

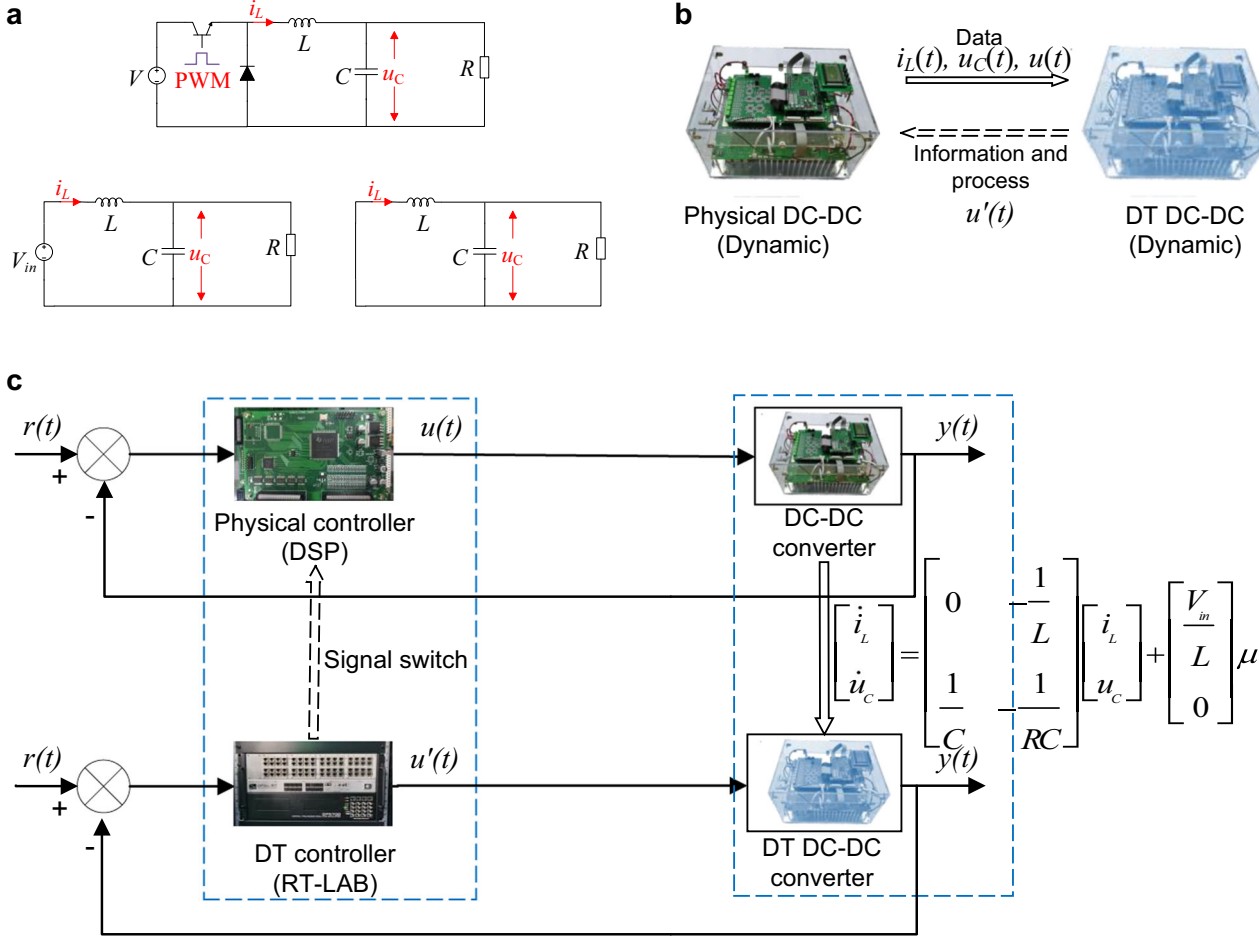

**Fig. 8 | DC-DC converter. a** Circuit diagram of the buck converter and the corresponding ON/OFF state. **b** Physical and twining DC-DC converter system. **c** Control architecture of the physical-DT DC-DC converter.

## Data availability

Source data are provided with this paper.

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

## Acknowledgements
This work was supported in part by the National Natural Science Foundation of China under Grant 62103308 (Z.L.), Grant 62073247 (W.H.), Grant 62173255 (G.L.), and Grant 62188101 (G.L.), in part by the Fundamental Research Funds for the Central Universities under Grant 2042023kf0095 (Z.L.), and in part by the China Postdoctoral Science Foundation under Grant 2022T150496 (Z.L.).

## Author contributions
Z.L. conceived the idea and wrote the manuscript. X.D. performed the simulation and experiments. Z.L. and X.D. analyzed data and interpreted the results. Funding was acquired by Z.L., W.H., and G.L. H.Z. was project administrator. All authors reviewed and edited the manuscript.

## Competing interests
The authors declare no competing interests.
