## [Peer Review File · Nature Communications]

Digital twin based monitoring and control for DC-DC convertersEditorial Note: Parts of this Peer Review File have been redacted as indicated to remove third-party material where no permission to publish could be obtained.

REVIEWER COMMENTS

Reviewer #1 (Remarks to the Author):

The paper "Digital Twin based Monitoring and Control for DC-DC Converters" describes the use of a digital twin applied to DC/DC converters. The abstract comprehensively summarizes the content of the paper and a common thread is present throughout. The format, figures and tables are understandable and readable.

Overall, the paper fulfills the criteria of scientific research, nevertheless, certain aspects need to be improved in order to be considered for acceptance.

(1) In the introduction it is not correct to indicate that there are few works related to the digital twin and DC/DC converters, since it is one of the most studied topics within the context of Power Electronics and the digital twin.

(2) How do you ensure a correct synchronisation between the digital model and the real model? As this is extremely important in the application of the adaptive Kalman filter. Explain this in detail in section II.A.

(3) What is the difference between using the Kalman filter to estimate the parameters and fit the real model and a classical observer?

(4) In the section "Case II System Model Variation", reference is made to the fact that the traditional mechanism does not represent the system because it would be necessary to modify the control matrix, however, a representation in state matrices considering as input to the system the V_{in} input voltage would be able to faithfully represent the dynamic behaviour of the system. Keep this in mind when justifying the case in question, you should rethink this case or at least indicate that the traditional method can meet this requirement.

(5) A more detailed description of possible cases that would justify the failure of the controller and thus the need to migrate to the digital controller that controls the digital twin should be included.

(6) Some information about the time step used to run the digital twin and the computational cost of the Kalman filter should be included.

A revision of the submission is mandatory.

Reviewer #2 (Remarks to the Author):

Dear Authors,
the topic of your research work is very timing and of interest for readers.

However, in my opinion, there are several issues to be solved in your manuscript.

1) You are proposing the DT concept for the application shown in Fig. 1; however, typical power values for applications as in Fig. 1, are not compatible with 48Vdc distribution grid. The current value would be too high to assure reasonable performances in terms of efficiency and drop voltage.

2) Fig.5 a, b, c are achieved by considering an IGBT. Fig. 5d is instead achieved with SiC MOSFET, however, the time scale is different and it is difficult to understand if there is a different behavior with respect to Fig. 5c.

3) Your sentence "the traditional mechanism model that is built based on a fixed input voltage V_{in} is no longer accurate" is only partially correct. In fact, also in the traditional mechanism model the input voltage V_{in} could be changed representing the actual V_{in} value, on the basis of the

generating unit operating conditions. Your comparison between the traditional model and the DT is not fair, at least in my opinion.

4) Which is the reason to change voltage reference in dc voltage distribution grid, like the one of your Fig. 1? It is quite a non-sense with respect to the proposed application.

5) Replacement of the physical controller by using the DT controller is possible only if the DT device is close enough to the physical converter. This is not the case in the majority of applications of interest for the DT concept. In fact the DT concept is of particular interest in either high power applications or complex systems and, usually, the DT device is far from the physical system.

Best regards

Reviewer #3 (Remarks to the Author):

The authors of the paper have presented and discussed the simulation and experimental results of a digital twin (DT) monitoring and control of a buck converter. The results showed that the controller of the DT is able to follow the physical controller response well during input and load variations, which makes it possible to swap controller to that of the DT when a failure occurs in the physical controller. The work on DT power electronics converter such as the buck converter is relevant and interesting at this time.

Below are comments/suggestions that need to be addressed before the paper can be accepted for publication:

- 1) Since the dynamic response comparisons are also carried with the so called, mechanism model-based system, the model description should also be provided in the paper for clarity
- 2) What is the type of optimization that have been employed in the DT controller?
- 3) Please provide details of the real-time DT resource (DSP, flip-flop and lookup-table where applicable) usage percentage and time results compared to the mechanism model-based system and to the other state-of-the art.
- 4) Parameter drift due to aging is common in dc-dc converters. Could the authors comment on how the DT response would be, to degradation in capacitor or MOSFET (increase in RDSon)?
- 5) In Section IV A, please clarify what are the special scenarios that should also be considered and why.
- 6) Experimental results using the SiC MOSFET in the physical dc-dc converter should also be provided for completeness.

Authors' Responses to the Reviews of NCOMMS-23-11244

We would like to thank the three anonymous reviewers very much for your valuable comments and suggestions on our paper, which helped us improve the paper. We have carefully considered those comments and suggestions, and thoroughly checked and revised the paper accordingly. For review convenience, the revised parts in the revised manuscript are highlighted and marked in the highlighted version. A clean version is also provided. The following is a detailed description about how we have addressed the reviewers' concerns in the revised manuscript. **The columns and pages referred to in this response letter are based on the highlighted version.**

Responses to Reviewer 1's comments

Q - The paper "Digital Twin based Monitoring and Control for DC-DC Converters" describes the use of a digital twin applied to DC/DC converters. The abstract comprehensively summarizes the content of the paper and a common thread is present throughout. The format, figures and tables are understandable and readable.

Overall, the paper fulfills the criteria of scientific research, nevertheless, certain aspects need to be improved in order to be considered for acceptance.

A - We sincerely appreciate the time and effort you dedicated to reviewing our manuscript. Your valuable comments and suggestions have helped us greatly improve the manuscript. We have carefully revised the manuscript to address your concerns. We hope the quality of this revision has been improved.

(1) In the introduction it is not correct to indicate that there are few works related to the digital twin and DC/DC converters, since it is one of the most studied topics within the context of Power Electronics and the digital twin.

A - Thank you for your valuable comment. We have revised and reorganized the sentences to be more precise. Previous DT studies have mainly focused on condition and health monitoring while the focus of this paper is more focused on control perspectives.

We have revised the sentence to be more precise as follows in Column 2, Page 1 of the revised manuscript.

“As a transforming technology, the DT is one of the most studied topics within the power electronics area [16], [17], and is mainly focused on condition and health monitoring. In [16], a survey was carried out to identify the concepts, applications, challenges, and trends of DTs used in energy conversion systems based on power electronics. In [17], a DT-based health indicator estimation method for the condition monitoring of DC-DC converters was proposed, which is

mainly used for monitoring the degradation level of the capacitor and MOSFET (metal-oxide-semiconductor field-effect transistor), the control perspective of which has not been considered. The application of DTs in DC-DC converters can potentially provide benefits such as health monitoring, fault diagnosing [18] in photovoltaic systems and detection of system abnormalities for buck converters [19].

For simplicity without loss of generality, this brief explores a DT-based buck converter system, mainly from control perspectives.”

(2) How do you ensure a correct synchronisation between the digital model and the real model? As this is extremely important in the application of the adaptive Kalman filter. Explain this in detail in section II.A.

A - Thank you for this crucial concern. Synchronization between the digital model and the real model is the key to the studies in this paper. Equation (4) reflects a dynamic DT model, combined with the Kalman filter that has good convergence, the state estimation ensures that the DT model can track the physical system in real-time. In this regard, the synchronization between the DT model and the real model can be achieved.

As presented in the original manuscript, “The physical system runs in real-time, while the DT can dynamically track the physical system. In this scenario, the physical system sends the data including i_L , u_C , and μ to its digital counterpart, based on which the DT updates its model and parameters. This also suits for variations of the system model. For a mechanism model that is built based on a fixed system offline, the model can be inaccurate once the system parameters change. While for a DT model in this brief, it is updated online according to the real-time data from the physical system. Synchronization between the DT and the physical system can be ensured even if the physical model changes or parameters drift due to aging.”

We have added explanations in Column 1, Page 3 in the revised manuscript as follows:

“The Kalman filter adaptation algorithm [23], which possesses several advantages, such as the ability to reduce noise, low computational complexity, good convergence [24], and high time efficiency, can be used to address this issue.”

References:

[23] L. Ljung, *System Identification: Theory for the User*. Upper Saddle River, NJ: Prentice-Hall PTR, 1999.

[24] M. Li, R. Kang, D. T. Branson, and J. S. Dai, “Model-free control for continuum robots based on an adaptive Kalman filter,” *IEEE/ASME Trans. Mechatronics*, vol. 23, no. 1, pp. 286–297, 2018.

(3) What is the difference between using the Kalman filter to estimate the parameters and fit the real model and a classical observer?

A - To illustrate the main difference between using the Kalman filter and a classical observer, we have added the following descriptions in Column 1, Page 3 in the revised manuscript:

“Different approaches can be employed for state estimation, for example, Kalman filters and observers, however, it has been proved that the Kalman filter provides better estimation performance for states and exhibits satisfactory estimation accuracy in the presence of noise [22].”

Reference:

[22] J. Hui and J. Yuan, “Kalman filter, particle filter, and extended state observer for linear state estimation under perturbation (or noise) of MHTGR,” *Prog. Nucl. Energy*, vol. 148, p. 104231, 2022.

(4) In the section "Case II System Model Variation", reference is made to the fact that the traditional mechanism does not represent the system because it would be necessary to modify the control matrix, however, a representation in state matrices considering as input to the system the V_{in} input voltage would be able to faithfully represent the dynamic behaviour of the system. Keep

this in mind when justifying the case in question, you should rethink this case or at least indicate that the traditional method can meet this requirement.

A - Thank you for your valuable comments. If we understand your comments correctly, you mean the input voltage can be put outside the control matrix, like the following:

$$\begin{bmatrix} \dot{i}_L \\ \dot{u}_C \end{bmatrix} = \begin{bmatrix} 0 & -\frac{1}{L} \\ \frac{1}{C} & -\frac{1}{RC} \end{bmatrix} \begin{bmatrix} i_L \\ u_C \end{bmatrix} + \begin{bmatrix} \frac{1}{L} \\ 0 \end{bmatrix} (V_{in} * \mu)$$

From a mathematical perspective, this is correct. From an engineering and control perspective, this is also possible when the source provides a stable input voltage, for example, the source is a battery or a regulated power supply. However, as discussed in Section III.B (Case II System Model Variation) of the original manuscript, we considered a renewable system:

“For a renewable system, due to the intermittent feature, the input voltage may vary from time to time.”

For a renewable system, V_{in} is often obtained through a data-driven method while modeling, and it is not stable due to the intermittent feature of renewable systems, which means V_{in} is not controllable. Therefore, V_{in} that needs to be identified is put in the control matrix. While the duty cycle μ is known and controllable, thus, it is chosen as the control output in our method.

Following your comments, to make it clear, we have revised the description where V_{in} first appears in Column 1, Page 2 of the revised manuscript.

“ V_{in} represents the dc source input (not a constant for renewable energy systems due to the intermittent feature)”

(5) A more detailed description of possible cases that would justify the failure of the controller and thus the need to migrate to the digital controller that controls the digital twin should be included.

A - Thank you for your comment. Controller failure that is frequently encountered due to various reasons can cause system malfunctioning, and the DT controller that is constructed based on hybrid twin modeling and runs in the RT-LAB will be activated in a very short time and works as the controller to ensure a functioning system. The DT controller can be regarded as a redundant system to improve the stability and reliability of the physical system. As a cost-effective option, once controller failure has been detected and the RT-LAB that works as a DT controller is activated, a new DSP controller can be used to replace the failed controller at appropriate timing.

In Section IV-C of the original and revised manuscript, the following paragraph highlighted the importance of “the need to migrate to the digital controller”:

“As shown in Fig. 9, once the control input $\mu = 0$ has been detected, which means that the physical controller of the physical buck converter system fails, the DT controller that is constructed based on hybrid twin modeling and runs in the RT-LAB will be activated in a very short time and works as the controller to ensure a functioning system. As a cost-effective option, once controller failure has been detected and the RT-LAB that works as a DT controller is activated, a new DSP controller can be used to replace the failed controller at appropriate timing.”

We have added the following sentence in the above paragraph to further enhance the need to migrate to the digital controller, in Column 1, Page 6 of the revised manuscript:

“The DT controller can be regarded as a redundant system to improve the reliability and stability of the physical system.”

We have also added descriptions to justify the failure of the controller and thus the need to migrate to the digital controller in Page 4 of the revised manuscript as follows:

“For practical applications, controller failure that is frequently encountered due to various reasons can cause system malfunctioning [35], and may cause failures and damage to the entire system, which should be seriously considered and addressed to ensure reliable control. For power electronics applications, controller failure poses a significant challenge. For a photovoltaic power plant, down (not functional) probability is predominantly caused by controller and inverter failures [36]. Power converter and control circuit defects, such as sensor failure and controller failure, contribute to around 90% of electric drive failures [37], [38].”

References:

- [35] X.-M. Sun, G.-P. Liu, D. Rees, and W. Wang, “Stability of systems with controller failure and time-varying delay,” *IEEE Trans. Autom. Control*, vol. 53, no. 10, pp. 2391–2396, 2008.
- [36] N. Shahidirad, M. Niroomand, and R.-A. Hooshmand, “Investigation of PV power plant structures based on Monte Carlo reliability and economic analysis,” *IEEE J. Photovolt.*, vol. 8, no. 3, pp. 825–833, 2018.
- [37] Y. Song and B. Wang, “Survey on reliability of power electronic systems,” *IEEE Trans. Power Electron.*, vol. 28, no. 1, pp. 591–604, 2013.
- [38] A. Joseph, K. Desingu, R. Semwal, T. R. Chelliah, and D. Khare, “Dynamic performance of pumping mode of 250 MW variable speed hydro-generating unit subjected to power and control circuit faults,” *IEEE Trans. Energy Convers.*, vol. 33, no. 1, pp. 430–441, 2018.

(6) Some information about the time step used to run the digital twin and the computational cost of the Kalman filter should be included.

A - We have included the time step and computational cost of the Kalman filter in the revised manuscript. The time step is included in the revised manuscript as follows:

In Column 2, Page 3 for the simulation:

“The time step employed for running the DT is 0.1 ms.”

And in Column 2, Page 4 for the experimental verification:

“The time step used for running the DT is 0.1 ms, which is aligned with the simulation time step.”

The computational cost of the Kalman filter is included in Column 1, Page 3 in the revised manuscript as follows:

“The computational cost of the Kalman-filter-based estimation is closely linked to the number of estimated state variables [25]. For the case of the model with two estimated variables in this brief, the computational cost is relatively low. For complex nonlinear models and systems with more state variables, the feedback linearization technique can be an effective solution for reducing complexity [1].”

Reference:

[25] M. Raitoharju and R. Piche, “On computational complexity reduction methods for Kalman filter extensions,” *IEEE Aerosp. Electron. Syst. Mag.*, vol. 34, no. 10, pp. 2–19, 2019.

A revision of the submission is mandatory.

A - We have carefully revised the manuscript following your valuable comments and suggestions.

Responses to Reviewer 2's comments

Q - Dear Authors, the topic of your research work is very timing and of interest for readers.

A - Thank you for your positive comment. Your valuable comments and suggestions have helped us a lot in the revision of this manuscript.

However, in my opinion, there are several issues to be solved in your manuscript.

Q - 1) You are proposing the DT concept for the application shown in Fig. 1; however, typical power values for applications as in Fig. 1, are not compatible with 48Vdc distribution grid. The current value would be too high to assure reasonable performances in terms of efficiency and drop voltage.

A - Thank you for your valuable comments. In Fig. 1, we tried to demonstrate that the DC-DC converter plays a vital role in renewable energy systems, which highlights the importance of research studies on DC-DC converters.

48 V is a type of low-voltage dc microgrids that has been widely used for residential and communication applications [26], [27] and for simulation and experimental verification in microgrids and renewable energy studies [28], [29].

The proposed DT system can be applied to different scenarios and different voltage levels, for example, to verify theories in converter-based microgrids. The simulation and experimental verification cases provided in our paper demonstrated the effectiveness of our proposed system.

We have conducted another simulation for 120 Vdc, as shown in Figure R1, which demonstrated that our proposed DT system and method can be applied to other voltage levels.

Figure R1: Simulation results for 120 Vdc.

We have revised the manuscript in Column 2, Page 3 for clarification as follows:

“The parameters for the simulation are listed in Table I, in which 48 V is a type of low-voltage dc microgrids that has been widely used for residential and communication applications [26], [27] and for simulation and experimental verification in microgrids and renewable energy studies [28], [29].”

Q - 2) Fig.5 a, b, c are achieved by considering an IGBT. Fig. 5d is instead achieved with SiC MOSFET, however, the time scale is different and it is difficult to understand if there is a different behavior with respect to Fig. 5c.

A - Thank you for your careful review. We added Fig. 5d to illustrate the generalization ability for both IGBT and SiC in the original manuscript and we apologize that we forgot about the time scale. In the revised manuscript, we have changed the time scale in Fig. 5d to be consistent with Fig. 5 a, b, c. The revised Fig.5d provides a simulation case similar to Fig.5c, where controller failure is detected and replaced by the DT controller to ensure a functioning system.

Fig. 5(d) SiC MOSFET-based simulation

Fig. 5. Simulation results of the four cases. (a) The output voltage when the reference voltage is changed from 0 V to 48 V and from 48 V to 20 V, (b) the output voltage when the input voltage is changed from 100 V to 90 V, and then from 90 V to 80 V, (c) the output voltage when controller failure is detected and replaced by the DT controller to ensure a functioning system, and (d) SiC MOSFET is used as the switch in the circuit as the case in (c).

Q - 3) Your sentence "the traditional mechanism model that is built based on a fixed input voltage V_{in} is no longer accurate" is only partially correct. In fact, also in the traditional mechanism model the input voltage V_{in} could be changed representing the actual V_{in} value, on the basis of the generating unit operating conditions. Your comparison between the traditional model and the DT is not fair, at least in my opinion.

A - Thank you for your valuable comments. As the response to Reviewer 1, if we understand your comments correctly, you mean the input voltage can be put outside the control matrix, like the following:

$$\begin{bmatrix} \dot{i}_L \\ \dot{u}_C \end{bmatrix} = \begin{bmatrix} 0 & -\frac{1}{L} \\ \frac{1}{C} & -\frac{1}{RC} \end{bmatrix} \begin{bmatrix} i_L \\ u_C \end{bmatrix} + \begin{bmatrix} \frac{1}{L} \\ 0 \end{bmatrix} (V_{in} * \mu)$$

From a mathematical perspective, this is correct. From an engineering and control perspective, this is also possible when the source provides a stable input voltage, for example, the source is a battery or a regulated power supply. However, as discussed in Section III.B (Case II System Model Variation) of the original manuscript, we considered a renewable system:

“For a renewable system, due to the intermittent feature, the input voltage may vary from time to time.”

For a renewable system, V_{in} is often obtained through a data-driven method while modeling, and it is not stable due to the intermittent feature of renewable systems, which means V_{in} is not controllable. Therefore, V_{in} that needs to be identified is put in the control matrix. While the duty cycle μ is known and controllable, thus, it is chosen as the control output in our method.

Following your comments, to make it clear, we have revised the description where V_{in} first appears in Column 1, Page 2 of the revised manuscript.

“ V_{in} represents the dc source input (not a constant for renewable energy systems due to the intermittent feature)”

Q - 4) Which is the reason to change voltage reference in dc voltage distribution grid, like the one of your Fig. 1? It is quite a non-sense with respect to the proposed application.

A - Fig. 1 is an example of DC-DC converters in renewable energy systems. In this application, the voltage reference may not need to be changed. We consider more application scenarios rather than the dc voltage distribution grid, for example, as discussed in Section III in Column 1, Page 4 of the original manuscript,

“Although 48 V is considered in this brief, different output voltages may be needed in other scenarios. For example, 20 V was used for charging batteries [22].”

In industry, especially in energy related applications, voltage reference changes are commonly used to test the tracking ability of the proposed system. For example, in [30] and [31]. Moreover, for a control-oriented DT system, voltage reference changes, normally, a step-up change and/or a step-down change, are typical to test tracking performance, such as in other control systems [32] and [33].

We have revised the manuscript to include more explanations on voltage reference change cases in Column 2, Page 3 in the revised manuscript.

“In industry, especially in energy-related applications, voltage reference changes are commonly employed to evaluate the tracking capability of a proposed system [30], [31]. Moreover, in the context of control-oriented research studies, it is prevalent to perform voltage reference changes, typically in the form of step-up or step-down adjustments, to assess the tracking performance [32], [33]. Therefore, reference tracking, as a basic control task, is first verified in this case.”

References:

- [30] M. S. Sadabadi, N. Mijatovic, J.-F. Tregouet, and T. Dragicevic, “Distributed control of parallel dc–dc converters under fdi attacks on actuators,” *IEEE Trans. Ind. Electron.*, vol. 69, no. 10, pp. 10 478– 10 488, 2022.

- [31] S. Zhuo, A. Gaillard, L. Xu, H. Bai, D. Paire, and F. Gao, "Enhanced robust control of a dc-dc converter for fuel cell application based on high-order extended state observer," *IEEE Trans. Transport. Electric.*, vol. 6, no. 1, pp. 278–287, 2020.
- [32] G.-P. Liu, "Tracking control of multi-agent systems using a networked predictive PID tracking scheme," *IEEE/CAA J. Autom. Sin.*, vol. 10, no. 1, pp. 216–225, 2023.
- [33] G.-P. Liu, "Coordinated control of networked multiagent systems via distributed cloud computing using multistep state predictors," *IEEE Trans. Cybern.*, vol. 52, no. 2, pp. 810–820, 2022.

Q - 5) Replacement of the physical controller by using the DT controller is possible only if the DT device is close enough to the physical converter. This is not the case in the majority of applications of interest for the DT concept. In fact the DT concept is of particular interest in either high power applications or complex systems and, usually, the DT device is far from the physical system.

A - Thank you for your valuable comments. Indeed, the replacement of the physical controller using a DT controller requires the DT controller to behave like the physical controller, which means it should resemble the physical converter closely enough. As far as we are concerned, the crux of DT and the usefulness of DT in real scenarios is to mimic the physical system as closely as possible.

In our application, the DT model is dynamically adapted to the physical system. Equation (4) reflects a dynamic DT model, combined with the Kalman filter that has good convergence, the state estimation ensures that the DT model can track the physical system in real-time.

As presented in the original manuscript, "The physical system runs in real-time, while the DT can dynamically track the physical system. In this scenario, the physical system sends the data including i_L , u_C , and μ to its digital counterpart, based on which the DT updates its model and parameters. This also suits for variations of the system model. For a mechanism model that is built based on a fixed system offline, the model can be inaccurate once the system parameters change.

While for a DT model in this brief, it is updated online according to the real-time data from the physical system. Synchronization between the DT and the physical system can be ensured even if the physical model changes or parameters drift due to aging.”

In this sense, we believe our DT device is close enough to the physical converter, thus, replacement of the physical controller by using the DT controller is possible.

Responses to Reviewer 3's comments

Q - The authors of the paper have presented and discussed the simulation and experimental results of a digital twin (DT) monitoring and control of a buck converter. The results showed that the controller of the DT is able to follow the physical controller response well during input and load variations, which makes it possible to swap controller to that of the DT when a failure occurs in the physical controller. The work on DT power electronics converter such as the buck converter is relevant and interesting at this time.

A - Thank you for your valuable time, effort, and comments. We have further revised the manuscript to address all your concerns.

Below are comments/suggestions that need to be addressed before the paper can be accepted for publication:

Q - 1) Since the dynamic response comparisons are also carried with the so called, mechanism model-based system, the model description should also be provided in the paper for clarity

A - Thank you for your comment. Descriptions of the mechanism model-based system have been added in Column 1, Page 2 in the revised manuscript as follows:

“A typical DC-DC buck circuit is illustrated in Fig. 2, along with its equivalent circuit during the ON and OFF states. Within one switching cycle of the switch, the system dynamic equations can be derived based on Kirchhoff's laws [21]. During the ON state of the switch, the inductor current i_L and the capacitor voltage u_C can be expressed as follows.

$$\begin{aligned} i_L &= C \frac{du_C}{dt} + \frac{u_C}{R}, \\ u_C &= V_{in} - L \frac{di_L}{dt}, \end{aligned} \tag{1}$$

where V_{in} represents the dc source input (not a constant for renewable energy systems due to the intermittent feature), and L , C , and R represent the inductance of the inductor, the capacitance of

the capacitor, and the load resistance, respectively. During the OFF state of the switch, their state equations are given by:

$$\begin{aligned} i_L &= C \frac{du_C}{dt} + \frac{u_C}{R}, \\ u_C &= -L \frac{di_L}{dt}. \end{aligned} \quad (2)$$

Let the duty cycle be defined as $\mu \in (0,1)$, which is also the control input.”

Fig. 2. Circuit diagram of the buck converter. (a) Fundamental circuit, (b) ON state, and (c) OFF state.

This model has been proven to effectively reflect the dynamic/steady-state characteristics of the system by [21], as shown in Figure R2 (from [21]).

[REDACTED]

Figure R2: Dynamic/steady-state characteristics of the mechanism model-based system.

Q - 2) What is the type of optimization that have been employed in the DT controller?

A - We used a Kalman filter to obtain the dynamic DT model to ensure an accurate model of the DT. As for the DT controller, in our study, it was designed offline using classical control theory. However, our previous work focused on online optimization of control parameters to achieve improved control performance, as referenced in [R1] and [R2]. Nevertheless, we have not extended this approach to the digital twin domain addressed in our current research, which presents both a challenging and promising direction for our future research efforts.

We have added the following sentence at the end of the Conclusion and Future Work section:

“DT controller optimization is also a challenging and promising direction for future work.”

References:

[R1] Liu, G.P. and S. Daley, Optimal-Tuning PID Control for Industrial Systems, *Control Engineering Practice*, vol. 9, no. 11, pp. 1185-1194, 2001.

[R2] Liu, G.P. and S. Daley, Optimal-tuning nonlinear PID control for hydraulic systems, *Control Engineering Practice*, vol. 8, no. 9, pp. 1045-1053, 2000.

Q - 3) Please provide details of the real-time DT resource (DSP, flip-flop and lookup-table where applicable) usage percentage and time results compared to the mechanism model-based system and to the other state-of-the art.

A - Our digital twin model was not run on a DSP (Digital Signal Processor). Instead, for the convenience of controller design, we opted to use the real-time control platform RT-Lab. To address your inquiry regarding computational resources, we have experimentally validated the algorithm's cost by providing a table showcasing the real-time resource utilization, as shown in Table R1.

Table R1 Real-time resource utilization

	Mechanism Model	Digital Twin Model
CPU Usage	1.56 %	1.97 %

It is noted that the real-time controller RT-Lab also includes other functional components, which prevents us from quantitatively comparing the computational resource utilization of our approach against the traditional method. Nonetheless, with a sampling period of 0.1 ms, we can observe an approximate 26% increase in computational resource usage with our approach. Compared with the significant improvement in the function of the DT controller (mainly including the real-time update model, controller switching, etc.), this part of the increase in computing cost is worthwhile.

Furthermore, from a theoretical standpoint, we have added a description of the algorithm complexity in the revised manuscript, specifically in Column 1, Page 3.

“The computational cost of the Kalman-filter-based estimation is closely linked to the number of estimated state variables [25]. For the case of the model with two estimated variables in this brief, the computational cost is relatively low. For complex nonlinear models and systems with more state variables, the feedback linearization technique can be an effective solution for reducing complexity [1].”

Q - 4) Parameter drift due to aging is common in dc-dc converters. Could the authors comment on how the DT response would be, to degradation in capacitor or MOSFET (increase in RDSon)?

A - Thank you for your comment. One of the essences of using the Kalman filter for modeling is to address parameter drift, which is one advantage of our proposed method. We pointed out in Column 2, Page 2 of our original manuscript that:

“Based on circuit theory, the process knowledge/mechanism modeling can be used as follows. This model is easy to acquire and comprehend. However, **it suffers from parameter drift issues**, which means that once the system model of the circuit system changes, the model has to be

reconstructed from scratch. To solve this issue, a mechanism-data hybrid modeling method is adopted for the design and implementation of the DT model.”

By using the proposed modeling method of this paper, we can not only monitor the status of the physical system, but also control it if necessary. We have added sentences in Column 1, Page 3 in the revised manuscript to make it clearer as follows:

“It is crucial to emphasize that (4) represents a time-variant system, capable of accurately reflecting the parameters of the physical system. This indicates that the proposed system remains applicable and effective, even in the presence of parameter drift, such as degradation in capacitor or MOSFETs.”

Q - 5) In Section IV A, please clarify what are the special scenarios that should also be considered and why.

A - Thank you for your comments. We have revised the manuscript to include more explanations on voltage reference change cases in Column 2, Page 3 in the revised manuscript as follows:

“In industry, especially in energy-related applications, voltage reference changes are commonly employed to evaluate the tracking capability of a proposed system [30], [31]. Moreover, in the context of control-oriented research studies, it is prevalent to perform voltage reference changes, typically in the form of step-up or step-down adjustments, to assess the tracking performance [32], [33]. Therefore, reference tracking, as a basic control task, is first verified in this case.”

References:

[30] M. S. Sadabadi, N. Mijatovic, J.-F. Tregouet, and T. Dragicevic, “Distributed control of parallel dc–dc converters under fdi attacks on actuators,” *IEEE Trans. Ind. Electron.*, vol. 69, no. 10, pp. 10 478– 10 488, 2022.

- [31] S. Zhuo, A. Gaillard, L. Xu, H. Bai, D. Paire, and F. Gao, “Enhanced robust control of a dc–dc converter for fuel cell application based on high-order extended state observer,” *IEEE Trans. Transport. Electric.*, vol. 6, no. 1, pp. 278–287, 2020.
- [32] G.-P. Liu, “Tracking control of multi-agent systems using a networked predictive PID tracking scheme,” *IEEE/CAA J. Autom. Sin.*, vol. 10, no. 1, pp. 216–225, 2023.
- [33] G.-P. Liu, “Coordinated control of networked multiagent systems via distributed cloud computing using multistep state predictors,” *IEEE Trans. Cybern.*, vol. 52, no. 2, pp. 810–820, 2022.

Q - 6) Experimental results using the SiC MOSFET in the physical dc-dc converter should also be provided for completeness.

A - Thank you for your comment. We have developed a SiC MOSFET-based converter system to test the performance for completeness, as shown in Figure R3. The used IGBT and SiC MOSFET (NTH4L040N120SC1) have been included in Fig. 6 of the revised manuscript.

Figure R3: Developed SiC MOSFET based converter system and the SiC MOSFET (The SiC MOSFET is added in Fig. 6 as shown on the left side)

The experimental results of the SiC MOSFET are illustrated in Fig. 10. A subsection (D. Experimental Results with SiC MOSFET) has been added in Section IV Experimental Verification to discuss the Experimental results using the SiC MOSFET in the physical dc-dc converter in

Column 1, Page 6 of the revised manuscript as follows:

“D. Experimental Results with SiC MOSFET

To further verify the generalization ability of the proposed system and validate the results obtained from the simulation case in Section III-D, experimental verification regarding the SiC MOSFET is also conducted with a new set of buck converter. Owing to the adoption of the SiC MOSFET, the new converter has a compact size compared to the IGBT-based counterpart, maintaining identical voltage and power levels.

The experimental results of the output voltage are shown in Fig. 10. In this case, the duty cycle is the same as that in Section IV-C. The transient state is observed as shown at the top of Fig. 10, which demonstrates a remarkable dynamic tracking performance. For controller failure, the results show that the system can quickly adapt to the DT controller, thus, ensuring a smooth transition and maintaining good control performance. It can be seen that the error of the output voltage between the physical system and the DT system is 1.0 V, which is lower than 1.6 V in Fig. 9.”

Fig. 10. Experimental results of the output voltage when physical DSP controller failure is detected and has been replaced by the DT controller to ensure a functioning system (using SiC MOSFET as the switch).

REVIEWERS' COMMENTS

Reviewer #3 (Remarks to the Author):

The authors have addressed and clarified all my comments. I would like to recommend the paper to be accepted for publication.

Authors' Responses to the Reviews of NCOMMS-23-11244A

We would like to thank the two anonymous reviewers very much for your valuable comments and suggestions on our paper, which helped us improve the paper. We have carefully considered those comments and suggestions, and thoroughly checked and revised the paper accordingly. For review convenience, the revised parts in the revised manuscript are highlighted and marked in the highlighted version. A clean version is also provided. The following is a detailed description about how we have addressed the reviewers' concerns in the revised manuscript.

Responses to Reviewer 3's comments

Q - The authors have addressed and clarified all my comments. I would like to recommend the paper to be accepted for publication.

A - We'd like to thank you again for the time and effort you dedicated to reviewing our manuscript. Your valuable comments and suggestions have helped us greatly improve the manuscript.

Responses to Reviewer 2's remaining comments

In checking your responses to Reviewer #2 comments we believe 2 points remain to be better described and clarified in your manuscript:

*** Reviewer 2's comment: "Your sentence "the traditional mechanism model that is built based on a fixed input voltage V_{in} is no longer accurate" is only partially correct. In fact, also in the traditional mechanism model the input voltage V_{in} could be changed representing the actual V_{in} value, on the basis of the generating unit operating conditions. Your comparison between the traditional model and the DT is not fair, at least in my opinion."*

>> Remaining point of concern: Your response remains vague with regards to how V_{in} is obtained ("data-driven methods" must be clearly defined). It remains unclear why V_{in} cannot be measured and used to update the "mechanism-based model" directly. You mention it being built offline, but it isn't clear why it could not be an online model.

A - Thank you for your comment. Firstly, regarding the "data-driven methods", we obtain V_{in} through an estimator based on the Kalman filter in our study. As it was mentioned earlier in the text, we did not reiterate this point in our previous response letter. Additionally, in our understanding, conventional mechanism-based models are generally fixed. Even if V_{in} is measurable and not included in the system matrix, the RLC components within the system might be time-varying, a scenario that traditional models struggle to address. Lastly, in renewable energy systems, voltage fluctuations are common due to variability on the generation side. Equipment manufacturers often prioritize the outputs of converters, and an excessive number of sensors could escalate costs. As a result, V_{in} might sometimes be unmeasurable, underscoring the importance of online parameter updates in the twin model.

In Pages 5-7 of our revised manuscript, we have strengthened the description of the source of V_{in} for clarity.

“Based on circuit theory, the process knowledge/mechanism modeling can be used. This model is easy to acquire and comprehend with actual measured values. However, it is generally fixed and thus suffers from parameter drift issues, which means that once the system model of the circuit system changes, the model has to be reconstructed from scratch. For example, V_{in} is typically fixed when using mechanism-based modeling owing to cost considerations. In addition, in renewable energy systems, voltage fluctuations (changes of V_{in}) are common due to variability on the generation side. Even if V_{in} is not fixed and measurable, the R, L and C components within a specific DC-DC system may be time-varying, which makes mechanism-based modeling challenging. To solve this issue, a mechanism-data hybrid modeling method is adopted for the design and implementation of the DT model.”

“ A_d and B_d (noting that V_{in} is included in B_d) are the system and control matrix discretized from (3)”

“As V_{in} is included in B_d (thus in θ), it is also updated in real time.”

>> Beyond this, it still isn't entirely clear in the manuscript if the Kalman filter is only applied to the digital twin or also to the mechanism-based model. If it's only to the DT model, please explain why. You may use this field to respond to these concerns, but please ensure they are also clarified within the main text.

A - Thank you for your comment. In this study, the Kalman filter is indeed only used for the DT model, while the parameters in the mechanism-based model are real measured values. This distinction arises from the unique characteristics and purposes of the two models. The DT model facilitates real-time analysis, monitoring, and control through live data, making the Kalman filter an advantageous choice due to its capacity to enhance state estimation accuracy and enable effective control. Conversely, the mechanism-based model aims to capture the fundamental physical principles and dynamics of the DC-DC converter. Although the application of the Kalman filter to this model is possible, it contradicts its original intent and aligns more closely with system identification traits. In the revision, we offer clearer explanations for the rationale behind using the

Kalman filter.

We have enhanced the manuscript on Page 7 as follows:

“It should be noted that the Kalman filter is only applied to the DT model, while the parameters in the mechanism-based model are actual measured values. The aim of the mechanism-based model is to accurately represent the fundamental physical principles and dynamics of the DC-DC converter. The DT model, on the other hand, facilitates real-time analysis, monitoring, and control through live data, making the Kalman filter an advantageous choice due to its capacity to enhance state estimation accuracy and enable effective control.”